# Genetic variation in promoter region of the bovine LAP3 gene associated with estimated breeding values of milk production traits and clinical mastitis in dairy cattle

**Destaw Worku**[¤]*, **Gopal Gowane, Archana Verma**

Animal Genetics and Breeding Division, ICAR-National Dairy Research Institute, Karnal, Haryana, India

¤ Current address: Department of Animal Science, College of Agriculture, Food and Climate Science, Injibara University, Injibara, Ethiopia
* destawworku@gmail.com

**Data Availability Statement:** All relevant data are within the paper and its Supporting Information files.

## Abstract

The purpose of this study was to identify genetic variants in the promoter and 5'UTR regions of bovine leucine amino peptidase three (LAP3) gene and analysed their associations with estimated breeding values (EBVs) of milk production traits and clinical mastitis in Sahiwal and Karan Fries cattle. Eleven SNPs were identified within the region under study of the LAP3 gene, including seven promoter variants (rs717156555: C>G, rs720373055: T>C, rs715189731: A>G, rs516876447: A>G, rs461857269: C>T, rs136548163: C>T, and rs720349928: G>A) and four 5'UTR variants (rs717884982: C>T, rs722359733: C>T, rs481631804: C>T and rs462932574: T>G). Out of them, 10 SNPs variants were found in both Sahiwal and Karan Fries cattle, with one SNP variant (rs481631804: C>T) being unique to Karan Fries cattle. Seven of these identified SNPs were chosen for association analyses. Individual SNP based association analysis revealed that two SNPs (rs720373055: T>C and rs720349928: G>A) were significantly associated with EBVs of lactation milk yield (LMY), 305-day milk yield (305dMY), and one significant association of SNP rs722359733: C>T with lactation length (LL) was observed. Haplotype based association analysis indicated that diplotypes are significantly associated with EBVs of LMY, 305dMY, and LL, individuals with H1H3 (CTACGCT/GCGTACG) being linked to higher lactation performance than other diplotypes. Further logistic regression analysis revealed that, animals with diplotype H1H3 was less susceptible to the incidence of clinical mastitis than other cows, as the odds ratio for the non-incidence of clinical mastitis was found to be low. Altogether, variations in the LAP3 gene promoter could be used as a genetic marker, most notably diplotype H1H3, may greatly benefit the simultaneous improvement of mastitis resistance and milk yield traits in dairy cattle. Moreover, bioinformatics analysis predicted that the SNPs rs720373055: T>C, rs715189731:A>G and rs720349928: G>A is located in the core promoter region and in TFBs, play key role in regulation of studied phenotypes.

**Funding:** This research was financially supported by Animal Genetics and Breeding Division of ICAR-National Dairy Research Institute, Karnal, India. Destaw Worku received the fund. The funder had no role in study design, data collection and analysis, decision to publish, or preparation of the manuscript.

**Competing interests:** The authors have declared that no competing interests exist.

## Introduction

Within India's economy, the dairy industry is regarded as one of the most important and dynamic agri-food sectors, and it serves as the primary source of income for dairy farmers. As a result, genetic improvement of milk production traits remains the primary economic activity for profitable dairy business. Sahiwal is one of the most important milch breeds of indigenous cattle famous for higher milk production, remarkable power of endurance for hot climate of subtropics, comparatively resistant to diseases and low maintenance cost [1]. Although the indigenous cows are less prone to mastitis, rigorous selection for more milk production has increased cow's susceptibility to mastitis [2, 3]. Whereas, Karan Fries is a synthetic breed evolved by crossing Holstein Friesian (75%) and Tharparkar cattle. It has good production potential but is more prone to mastitis as compared to their Indian counterpart, while the prevalence of clinical mastitis in crossbred cows ranges between 5 and 37% [4] and it is associated with many animal and environmental level factors [5]. For many years, general mastitis control strategies have met with little success. As a result, mastitis continues to remain the most challenging cattle disease with great economic implications through reduced quantity, quality, and waste of milk unfit for consumption, resulting in massive antibiotic use and premature culling, veterinary service, and labour cost [6, 7]. Due to the important economic value of milk production traits, a deeper understanding of genes involved in resistance to mastitis will take a step closer to the aim of developing cattle breeds, less susceptible to mastitis, whilst maintaining desired milk production traits. In this regard, marker-assisted selection (MAS) can be used to select possible genotypes associated with the most desirable phenotypes, which could enhance and expedite the efforts for genetic improvement.

Advances in the identification of loci and chromosomal regions that control economically important traits have opened opportunities for improving milk production traits and mastitis resistance in dairy cattle breeds. The candidate gene approach, which selects useful markers for mastitis resistance while maintaining production traits, is of utmost importance in dairy industry. Thus, studying the genetic variability of candidate genes and their associations with milk yield traits and mastitis is necessary [8, 9].

The bovine leucine amino peptidase three (LAP3) gene, which spans a 25-kilobase genomic segment and encodes a 519 amino-acid mature protein, is found on Bos taurus autosome 6 (BTA6). The LAP3 also known as PEPS gene is involved in the metabolism of many peptides, including hormone control, protein maturation, protein inactivation and protein digestion in the terminal stage [10, 11]. In mammals, the LAP3 gene is involved in the processing of bioactive peptides (oxytocin, vasopressin, and enkephalins) as well as vesicle trafficking to the plasma membrane for MHCI antigen presentation [12, 13]. So far, [14] showed that the LAP3 gene for the BTA6 QTL positional candidates is differentially expressed in bovine mammary tissue at different stages of lactation cycle (pregnancy, lactation, and involution). They noticed, a 2.2-fold increase in expression in lactating mammary gland tissues compared to tissue from late pregnancy. A QTL with a strong influence on milk production traits was also identified, and mapped to a 420-kb gap on bovine chromosome 6 between the genes ABCG2 and LAP3 [15–17]. As a result, Olsen and colleagues [18] identified a QTL in a 420-kb region of the genome that contains six milk production candidate genes, including the LAP3 gene, on bovine chromosome 6. On the other hand, it has been shown that some LAP3 gene variants are associated with economic traits in dairy cattle [19–22]. Furthermore, Ju et al. [23] discovered two genetic polymorphisms in the bovine PEPS gene promoter region (g.-534T>C and g.-2545G>A), and their combined haplotypes were linked to milk fat percentage and somatic cell score in Chinese Holstein cattle. Based on the evidence presented above, it is believed that the LAP3 gene could be a promising functional gene for milk production traits in dairy cattle.

Nonetheless, identifying single-nucleotide polymorphisms (SNPs) affecting milk production traits in tropical cattle is of paramount importance to accelerate the rate of genetic change in the developing world's dairy industry [24]. However, studies on the relationship between polymorphisms in bovine LAP3 gene with milk production traits and incidence of clinical mastitis in Sahiwal and Karan Fries cattle are still lacking. Therefore, the current study sought to identify SNPs located in the promoter region of the bovine LAP3 gene and their associations with estimated breeding values (EBVs) of milk production traits and clinical mastitis in Sahiwal and Karan-Fries cattle.

## Materials and methods

### Ethics statement

All the experimental animals were approved by the Institutional Animal Ethics Committee (IAEC) of ICAR-National Dairy Research Institute, Karnal, India, on 22-07-2020 under approval number 46-IAEC-2021. Moreover, the experiments were conducted in accordance with the guidelines of Committee for the Purpose of Control and Supervision of Experimentation in Animals (CPCSEA), Government of India (1705/GO/ac/13/CPCSEA).

### Animals, blood samples and phenotypic data

Blood samples were obtained from 220 cows, 110 Sahiwal and 110 Karan Fries, that were reared at ICAR-National Dairy Research Institute (ICAR-NDRI), Karnal, India, under similar managemental conditions. The traits used for the study were production traits such as total lactation milk yield (LMY), 305-day milk yield (305dMY), 305-day fat yield (305dFY), 305-day solid not fat yield (305dSNFY), lactation length (LL) and clinical mastitis (CM). Based on records of 935 Sahiwal cows (1979–2019) and 1426 Karan Fries cows (1988–2019) with all parities maintained in ICAR-NDRI, India, estimated breeding values (EBVs) for milk production traits were estimated by repeatability animal model using BLUPF90 software [25]. As a result, for the association analysis of SNPs in the LAP3 gene with milk production traits, the EBVs of genotyped 110 Sahiwal and 110 Karan Fries animals were used as responding variables. Furthermore, data on the incidence of CM was collected from the Animal Health Complex treatment register, and animals were classified as healthy or mastitis-affected based on the frequency of clinical mastitis.

### DNA extraction and PCR amplification

Genomic DNAs were extracted from whole blood samples by phenol-chloroform extraction method following standard procedures [26]. The quantity and quality of extracted DNAs were measured by NANODROP 2000 Spectrophotometer (Thermo Scientific, DE, USA). Samples showing an optical density (OD) ratio (260 nm/280 nm) between 1.7 and 1.9 were used for further analysis. The stock DNA was further diluted to the final concentration of 50 ng/µL in nuclease-free water and stored at 4˚C. Primer used to amplify the target promoter region was designed based on nucleotide sequence of the Bos- taurus LAP3 gene (ENSBTAG00000005989) using Primer3 Plus online software [27]. Primers LAP3- F (5'–GCTACGTGCAACCTTTCTCC–3') and LAP3- R (5'–CTCACCTTCGTCATGTCTGC– 3') were used to amplify 537 bp (430 bp of 5' flanking region to 107 bp of exon 1) of the bovine LAP3 gene. The PCR reactions were carried out in a total volume of 25 µl on a Thermo- Cycler (Bio- Rad T100) containing 2.0 µl genomic DNA (50 ng /µl), 0.5 µl (10 pM of each primer), 13.0 µl of 2X PCR Master Mix and 9.0 µl of nuclease free water. The PCR reaction cycling protocol encompassed initial denaturation at 95˚C for 3 min, followed by 34 cycles of 94˚C for 30 s, specific annealing

temperature (59.5˚C) for 30 s, 40 s at 72˚C, and a final extension step at 72˚C for 8 minutes. The PCR products were evaluated by 1.5% agarose gel electrophoresis by staining with ethidium bromide.

## SNP identification and genotyping

The representative samples of purified PCR products for both breeds were sequenced using the automated ABI3730XL DNA sequencer (Applied Biosystems, Foster city, CA, USA) to search for polymorphism. Sequenced data were analyzed using CodonCode Aligner [28], to discover, analyze the sequences and to find the mutation sites and its location. Multiple sequence alignments were performed with MEGA11 [29] software. After discovery of the SNP sites, all samples (110 Sahiwal and 110 Karan Fries = 220) were further genotyped using ABI3730XL DNA sequencer (Applied Biosystems, Foster city, CA, USA).

## Statistical analysis

The population genetics statistical analysis: genotypic and allelic frequencies, Hardy-Weinberg equilibrium test ($\chi^2$), polymorphism information content (PIC), gene heterozygosity (Ho and He) and effective allele numbers (ne) calculated using PopGen2 software [30]. Haplotype analysis among the identified SNPs of LAP3 gene was performed using the SNPstat online server [31].

Association analysis between each individual SNPs genotypes and haplotype combinations and the EBVs of milk production traits of 220 cows were carried out using a general linear model procedure of SAS Version 9.2 software [32] with the following linear model.

$$Y_{ijk} = \mu + B_i + G_j + e_{ijk}$$

where $Y_{ijk}$ = Estimated breeding value of milk production traits, $\mu$ is overall mean, $B_i$ is the effect of $i^{th}$ breed, $G_j$ is the fixed effect corresponding to the $j^{th}$ genotype or haplotype combination of individual and $e_{ijk}$ is the random error. EBVs were used as a phenotype to test the association of promoter variants of LAP3 gene with milk production traits. P values < 0.05 were regarded as significant. Tukey-Kramer multiple comparison tests were conducted to compare significant differences between groups. Furthermore, the additive (a), dominant (d), and substitution ($\alpha$) effects were calculated using the following formulas: $a = \frac{AA-BB}{2}$; $d = \frac{AB-AA+BB}{2}$; $\alpha$ = a+d (q-p) [33], where AA, BB, and AB are the least square means of EBVs for milk production traits in the respective genotypes, and p and q were the frequencies of allele A and B, respectively.

Haplotype based association with mastitis affected and non-affected animals were analyzed using the Chi-square ($\chi^2$) procedure of SAS Version 9.2 software [32]. Further logistic regression model was applied to analyze the fixed effect of breed, period of calving, season of calving and diplotypes or haplotype combination on the incidence of clinical mastitis by the maximum likelihood method of the Logistic procedure of SAS Version 9.2 software. The probability is modelled to the event of the incidence of clinical mastitis.

## Prediction of transcription factor binding sites (TFBs) and search for CpG islands

The Genomatix software suite [34] was used to predict whether the SNPs in the promoter region of LAP3 changed the TFBSs or not. To search for CpG, the stringent search criteria, Takai and Jones algorithm: GC content ≥ 50%, Observed CpG/Expected CpG ratio ≥ 0.65,

and length > 200 bp was used [35]. For this purpose, the EMBOSS Cpgplot available at web link https://www.ebi.ac.uk/Tools/seqstats/emboss_cpgplot/) was used [36].

## Results and discussion

### Identification of single nucleotide polymorphisms (SNPs)

Through sequencing (Fig 1), a total of 11 SNPs were identified in the promoter (rs717156555: C>G, rs720373055: T>C, rs715189731: A>G, rs516876447: A>G, rs461857269: C>T, rs136548163: C>T and rs720349928: G>A) and 5'UTR (rs717884982: C>T, rs722359733: C>T, rs481631804:C>T and rs462932574: T>G) of bovine LAP3 gene (Table 1). Of them, 10 SNPs variants were found in both Sahiwal and Karan Fries cattle, with 1 SNP variant (rs481631804: C>T) being unique to Karan Fries cattle. To the best of our knowledge, the eleven unique SNPs discovered in this study have never been reported in dairy cattle before. In Sahiwal and Karan Fries cattle populations, all of the detected SNPs showed polymorphism, however the SNP rs136548163: C>T tended to have a monomorphic pattern in Sahiwal cattle. It is worth nothing that the SNPs rs720373055: T>C and rs715189731: A>G were always linked in the study population in the following manner: (1) Animals homozygous at position rs720373055: T>C were always homozygous at position rs715189731: A>G, and heterozygotes were heterozygous at both sites; (2) the individuals had either T at position rs720373055: T>C, and A at rs715189731: A>G, or C at position rs720373055: T>C and G at rs715189731: A>G. As a result, it might be concluded that the two SNPs rs720373055: T>C and rs715189731: A>G were entirely linked and inherited as a unit in the studied populations tested. Among eleven SNPs being detected, seven of them, which are present in both the breeds with frequency more than 5% were chosen for association analysis. Details of the identified SNPs including, region and location in LAP3 gene are illustrated in Table 1.

### Allelic and genotypic frequencies of SNPs

Table 2 summarizes the genotypic and allelic frequencies, PIC, Ho, He, ne, and the Hardy-Weinberg equilibrium $\chi^2$ test for the identified SNPs of LAP3 gene promoter variants in Sahiwal and Karan Fries cattle. Except for the SNP rs516876447: A>G, where the mutant allele frequency was higher than the wild type allele frequency in the examined population, the wild type allele frequencies were the dominant alleles at loci for other SNPs. According to Chi squared test, for 3 SNP generated genotypes (rs717156555: C>G, rs516876447: A>G & rs722359733: C>T), the individual frequencies of the genotypes significantly differed from the Hardy-Weinberg equilibrium (P < 0.05). The possible explanation for this could be: (1) Sahiwal and Karan Fries cattle are among the well-established indigenous and crossbred cattle breeds in India known to have good genetic potential to produce considerably large quantities of milk, selection pressure for increased milk production infers the loss of non-favourable alleles. (2) The effect of small population size used for the study cannot be neglected. The PIC is a useful variable for assessing genetic diversity from different candidate gene loci. Our results showed that all SNPs were in medium polymorphism level (0.25<PIC<0.50). Furthermore, the population under study has a medium to high effective allele number (ranging from 1.42 to 1.97). This suggests that these SNPs may be subjected to selection.

### Haplotype analysis

Haplotype analysis was carried out using the SNPstat online server (https://www.snpstats.net/analyzer.php) [31]. Within the studied population, 16 haplotypes were detected (S1 Table), with frequencies ranging from 0.4272 (H1: CTACGCT) to 0.001(H16: CTGCGCG),

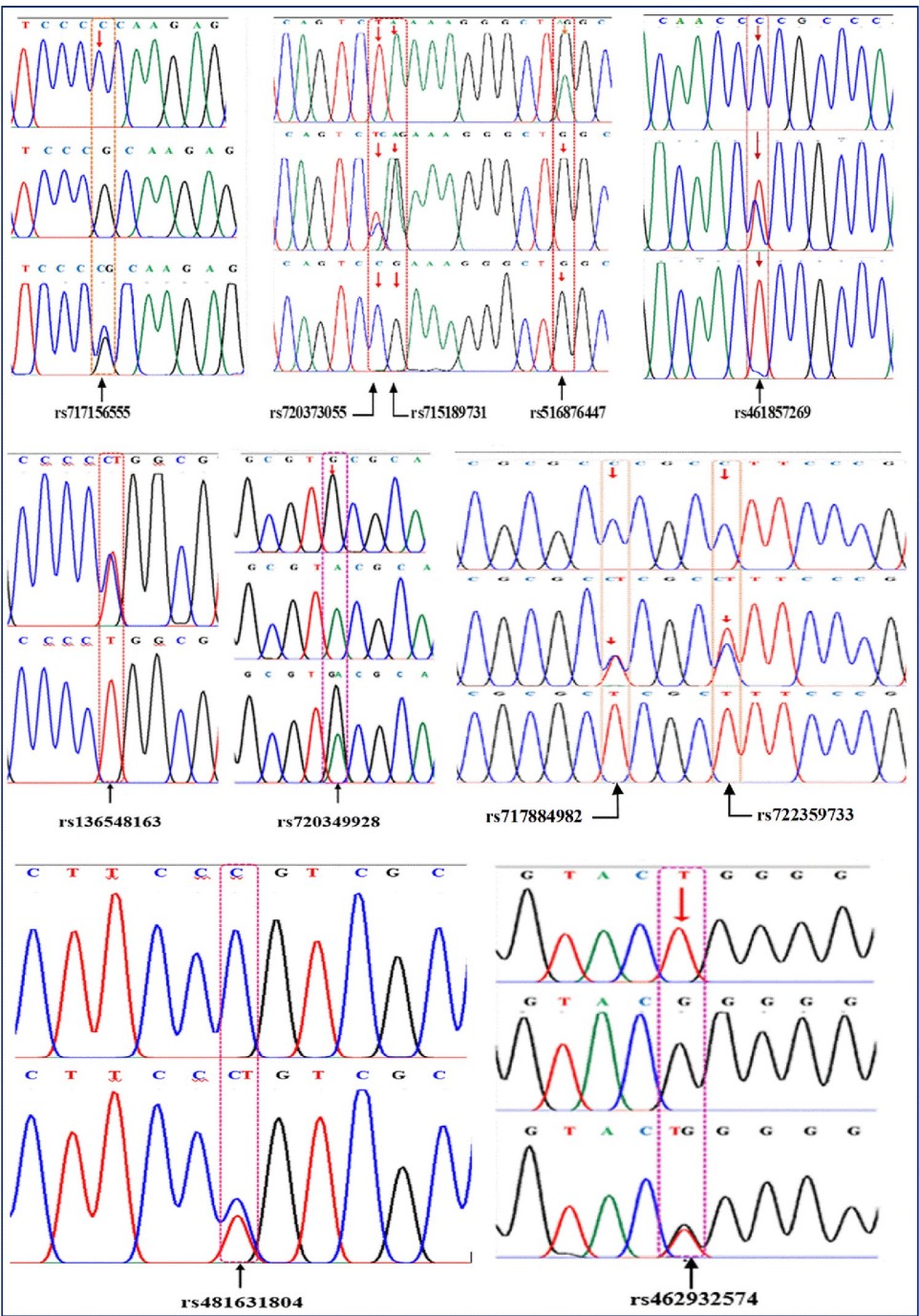

**Fig 1. Schematic representation of SNPs detected in promoter region of the bovine LAP3 gene.** The arrow indicates the position of mutation sites. The number indicates the single nucleotide polymorphisms identifier (SNP ID).

respectively. However, genetic polymorphisms prevalent in both populations and haplotypes with frequencies greater than 5% were kept for association analysis. As a result, in the association study, four haplotypes were used: H1(CTACGCT), H2 (CTGCGTT), H3 (GCGTACG), and H4 (CTGCGCT), as well as five diplotypes: H1H1(CTACGCT/ CTACGCT), H1H2

**Table 1. Single nucleotide polymorphisms (SNPs) detected by sequencing in the LAP3 gene (ARS–UCD1.2).**

| No. | SNP ID | Region in the gene | Location (BTA:bp) | Nucleotide substitution |
|---|---|---|---|---|
| 1 | rs717156555 | Promoter | 6:37140485 | C > G |
| 2 | rs720373055 | Promoter | 6:37140513 | T > C |
| 3 | rs715189731 | Promoter | 6:37140514 | A > G |
| 4 | rs516876447 | Promoter | 6:37140523 | A > G |
| 5 | rs461857269 | Promoter | 6:37140644 | C > T |
| 6 | rs136548163 | Promoter | 6:37140675 | C > T |
| 7 | rs720349928 | Promoter | 6:37140681 | G > A |
| 8 | rs717884982 | 5'UTR | 6:37140763 | C > T |
| 9 | rs722359733 | 5'UTR | 6:37140767 | C > T |
| 10 | rs481631804 | 5'UTR | 6:37140772 | C > T |
| 11 | rs462932574 | 5'UTR | 6:37140789 | T > G |

(CTACGCT/ CTGCGTT), H1H3 (CTACGCT/ GCGTACG), H1H4 (CTACGCT/ CTGCGCT) and H3H4 (GCGTACG/ CTGCGCT) (Table 4).

## Associations of SNPs and haplotype combinations with estimated breeding values of milk production traits

The associations analysis results between the SNPs in the promoter and 5'UTR of bovine LAP3 gene and EBVs of milk production traits are shown in Table 3. Two SNPs (rs720373055: T>C and rs720349928: G>A) were significantly associated with EBVs of LMY and 305dMY. In terms of LMY and 305dMY, animals with genotype CC (rs720373055: T>C) and GG (rs720349928: G>A) had significantly higher lactation performance (Table 3), indicating that allele C and G are associated with superior genetic merit for milk production traits in the studied population (S2 Table). There is also a significant association between the SNP

**Table 2. The genotypic and allelic frequencies, as well as population indices for the identified loci of LAP3 gene promoter variants in Sahiwal and Karan Fries cattle.**

| SNPs | Genotypic frequency | | | Allele frequency | | $\chi^2$ | PIC | Ho | He | ne* |
|---|---|---|---|---|---|---|---|---|---|---|
| rs717156555: C>G | CC | CG | GG | C | G | 4.70* | 0.27 | 0.269 | 0.316 | 1.459 |
| | 0.67 | 0.27 | 0.06 | 0.80 | 0.20 | | | | | |
| rs720373055: T>C | TT | TC | CC | T | C | 0.002 | 0.28 | 0.302 | 0.301 | 1.429 |
| | 0.64 | 0.29 | 0.07 | 0.78 | 0.22 | | | | | |
| rs516876447: A>G | AA | AG | GG | A | G | 18.475*** | 0.37 | 0.349 | 0.495 | 1.974 |
| | 0.27 | 0.35 | 0.38 | 0.44 | 0.56 | | | | | |
| rs461857269: C>T | CC | CT | TT | C | T | 1.088 | 0.26 | 0.288 | 0.310 | 1.447 |
| | 0.66 | 0.29 | 0.05 | 0.81 | 0.19 | | | | | |
| rs720349928: G>A | GG | GA | AA | G | A | 0.464 | 0.27 | 0.278 | 0.292 | 1.410 |
| | 0.66 | 0.28 | 0.06 | 0.80 | 0.20 | | | | | |
| rs722359733: C>T | CC | CT | TT | C | T | 6.660*** | 0.28 | 0.278 | 0.338 | 1.508 |
| | 0.65 | 0.28 | 0.07 | 0.79 | 0.21 | | | | | |
| rs462932574: T>G | TT | TG | GG | T | G | 0.017 | 0.26 | 0.307 | 0.304 | 1.435 |
| | 0.64 | 0.30 | 0.06 | 0.81 | 0.19 | | | | | |

*** = Significant at P < 0.001

* = Significant at P < 0.05, $\chi^2$ = Chi–square, PIC = Polymorphic information content, He = Expected heterozygosity, Ho = Observed heterozygosity, ne* = Effective number of alleles

**Table 3. Associations of different genotypes of SNPs in LAP3 gene with estimated breeding values of milk production traits.**

| Loci | Genotype | LMY (kg) | 305dMY (kg) | 305dFY (kg) | 305dSNFY (kg) | LL (day) |
|---|---|---|---|---|---|---|
| rs717156555: C>G | CC (142) | 17.92 | 143.60 | 2.14 | 2.68 | 4.29 |
| | CG (57) | 276.96 | 168.50 | -0.53 | -0.96 | 4.57 |
| | GG (13) | 202.96 | 90.02 | -1.74 | -2.85 | 5.62 |
| | P | 0.2742 | 0.7122 | 0.1345 | 0.3009 | 0.9862 |
| rs720373055: T>C | TT (136) | 1.26[b] | -141.62[b] | -2.41 | -4.91 | 1.87 |
| | TC (61) | -136.50[c] | -83.61[b] | -1.22 | -1.73 | -3.80 |
| | CC (15) | 633.08[a] | 627.35[a] | 3.48 | 5.51 | 16.41 |
| | P | 0.00614 | 0.0030 | 0.1437 | 0.1456 | 0.3763 |
| rs516876447: A>G | AA (57) | 185.98 | 138.33 | 0.19 | -0.07 | 7.16 |
| | AG (74) | 181.44 | 154.10 | 0.20 | 0.004 | 6.18 |
| | GG (81) | 130.42 | 109.67 | -0.53 | -1.06 | 1.15 |
| | P | 0.7351 | 0.6011 | 0.4808 | 0.6235 | 0.2242 |
| rs461857269: C>T | CC (141) | 173.05 | 213.35 | 0.55 | 1.30 | 9.04 |
| | CT (61) | 256.47 | 165.65 | -0.55 | -1.50 | 3.60 |
| | TT (10) | 68.32 | 23.12 | -0.15 | -0.94 | 1.84 |
| | P | 0.6395 | 0.4538 | 0.8202 | 0.6798 | 0.7631 |
| rs720349928: G>A | GG (140) | 401.20[a] | 260.66[a] | -0.58 | -0.800 | 0.62 |
| | GA (59) | 288.78[ab] | 305.14[a] | 2.45 | 3.39 | 16.62 |
| | AA (13) | -192.14[b] | -163.69[b] | -2.01 | -3.71 | -2.75 |
| | P | 0.0001 | 0.0001 | 0.0881 | 0.1380 | 0.1692 |
| rs722359733: C>T | CC (137) | 149.27 | 119.08 | -0.28 | -0.74 | 4.54[a] |
| | CT (59) | 51.72 | 46.84 | -0.66 | -1.28 | -4.68[b] |
| | TT (16) | 54.93 | 40.79 | -0.36 | -0.87 | -0.80[ab] |
| | P | 0.3551 | 0.2936 | 0.8045 | 0.8759 | 0.0096 |
| rs462932574: T>G | TT (135) | 55.01 | 44.78 | -0.16 | -0.61 | -0.42 |
| | TG (65) | 115.27 | 85.49 | -0.31 | -0.63 | 2.53 |
| | GG (12) | 85.65 | 76.45 | -0.83 | -1.66 | -3.04 |
| | P | 0.5011 | 0.5175 | 0.8364 | 0.8919 | 0.3232 |

[a,b,c] = Least squares means with different superscripts are significantly different (P < 0.05), LMY = Lactation milk yield, 305dMY = 305–day milk yield, 305dFY = 305–day fat yield, 305dSNFY = 305–day solid not fat yield, LL = Lactation length, kg = kilogram

rs722359733: C>T and LL, with genotype CC being linked to longer lactation than genotype CT and TT (P < 0.05), respectively. Furthermore, it was found that the promoter region rs720373055: T>C and rs715189731: A>G resulted in the generation of TFBSs for the LAP3 gene's corresponding transcription factor C2H2 zinc finger transcription factor 26 (ZF26). The SNP, rs720349928: G>A, was also predicted to alter the TFBSs for the corresponding transcription factors via zinc transcriptional regulatory element (ZTRE), CTCF and brother of the regulator of imprinted sites (BORIS) gene families, AHR-arnt heterodimers and AHR-related factors (AHRR), C2H2 zinc finger transcription factors 15 (ZF15) and Vertebrate homologues of enhancer of split complex (HESF), respectively. As a result, the higher lactation performance of CC (rs720373055: T>C) genotyped cows could be attributed to the regulation of rs720373055: T>C on the transcriptional activity of the LAP3 promoter, primarily through the transcription factor ZF26, where the T to C substitution may be responsible for changes in the animals' milk production parameters. Similarly, the presence of G allele at rs720349928: G>A created ZTRE, CTCF + BORIS gene family, AHRR, ZF15 and HESF transcription factor

**Table 4. Association of diplotypes of LAP3 gene with estimated breeding values of milk production traits in Sahiwal and Karan Fries cattle.**

| Diplotypes | Frequency (N) | LMY (kg) | 305dMY (kg) | 305dFY (kg) | 305dSNFY (kg) | LL (day) |
|---|---|---|---|---|---|---|
| H1H1 | 39.46 (58) | 88.14[b] | 61.88[cb] | -0.069 | -0.70 | 3.14[b] |
| H1H2 | 15.65 (23) | -16.83[c] | 9.12[c] | -0.56 | -1.22 | 0.44[b] |
| H1H3 | 17.01(25) | 353.87[a] | 289.29[a] | 1.33 | 2.09 | 10.92[a] |
| H1H4 | 17.01(25) | 99.72[ab] | 86.75[cb] | 0.26 | -0.22 | 4.32[b] |
| H3H4 | 10.88 (16) | 241.27[ab] | 233.65[ab] | 0.26 | 0.25 | 4.75[b] |
| | P | 0.0021 | 0.0001 | 0.3172 | 0.3056 | 0.0122 |

Least square means in the same column with different superscripts ([a], [b], and [c]) differ significantly at P < 0.01 and P < 0.001, H1H1 = CC–TT–AA–CC–GG–CC–TT, H1H2 = CC–TT–AG–CC–GG–CT–TT, H1H3 = CG–TC–AG–CT–GA–CC–TG, H1H4 = CC–TT–AG–CC–GG–CC–TT, H3H4 = CG–TC–GG–CT–GA–CC–TG. Haplotypes with frequency less than 0.05 were ignored in analysis.

binding sites, which was eliminated with the presence of the A allele could thus potentially affect LMY and 305dMY of AA genotype cows. However, further studies using gel shift assays are needed to confirm the regulatory role of these SNPs on milk production traits in cattle. Previous research; such as [37, 38], who reported that SNPs with in the regulatory regions can disrupt transcription via altering TFBSs or RNA stability. Wang et al. [39], noticed that SNPs in the transcription factor binding sites might cause phenotypic variation by inducing variances in gene expression.

Thus, the effect of haplotype combinations on EBVs milk production traits in cattle was analyzed (Table 4). The frequencies of haplotype combinations < 5.0% were not considered. Haplotypes composed of SNPs could provide accurate information than single SNP analysis for economic trait associations, due to the ancestral structure captured in the distribution of haplotypes [40]. In the current study, statistical analysis revealed a significant relationship between haplotype combinations and EBVs of LMY (P value = 0.0021), 305dMY (P value = 0.0001) and LL (P value = 0.0122), but not with 305dFY and 305dSNFY (Table 4). Cows carrying haplotype combination H1H3 (CTACGCT/ GCGTACG) had the highest EBVs of LMY, 305dMY and LL, whereas cows with H1H2 (CTACGCT/ CTGCGTT) had the lowest lactation performance. This study shows for the first time that the SNPs genotypes and haplotype combination in promoter variants of the LAP3 gene has a substantial correlation with EBVs of milk production traits in lactating dairy Sahiwal and Karan Fries cows.

Association between LAP3 gene and economically important traits in cattle was reported in a few studies [19–22]. The previous studies found that genetic changes in the promoter regions might result in large potential phenotype variability, which provides useful SNP marker information for dairy breeding schemes [39, 41]. Other than the SNPs loci identified in this study, [23] reported two novel SNPs in the promoter region of the bovine PEPS gene, and they found that the combined analysis of these SNPs, g.-534T>C and g.-2545G>A, were significantly associated with fat percentage and SCS traits in Chinese Holstein cattle. In addition, the mutation (g.25415T>C) within the 3'UTR (exon 13) region of the LAP3 gene was found to be statistically significant for protein percentage in three cattle breeds of China [20].

In light of all these, it is believed that QTL effects in the LAP3 gene on chromosome 6 altered milk production traits in the examined breeds. Previous studies [15, 16], showed the presence of one or more QTL for milk production traits on BTA6. A QTL with large impact on milk production traits was also mapped to a 420-kb interval on Bovine chromosome six between the genes ABCG2 and LAP3 [17]. Consequently, Olsen et al. [18] mapped a QTL in a 420- kb region of Bovine chromosome 6 that contains six milk production candidate genes, including the LAP3 gene. Despite the fact that the LAP3 gene, which is located near QTL, is

crucial for milk performance traits [18, 21, 22]. Another evidence claims that, statistically significant estimations of QTL impacts on breeding value in chromosome 6 of Holstein cattle varied from 340 to 640 kg of milk, 15.6 to 28.4 kg of fat, and 14.4 to 17.6 kg of protein [42]. In a nutshell, this study reported previously unknown SNPs in the promoter and 5'UTR regions of the LAP3 gene that were associated with EBVs of LMY, 305dMY and LL in lactating dairy cattle. According to the findings of this study, the bovine LAP3 gene could be a promising candidate gene for increasing milk production traits in dairy cattle, and it could be used in a marker assisted selection program to improve desirable traits in cattle. However, to form a firm conclusion on the effects of the significant SNPs genotypes and haplotype combinations, a substantial larger sample size is required.

## Associations between diplotypes and incidence of clinical mastitis

In this study, SNPs were evaluated individually and found to have no significant effects on the incidence of clinical mastitis. However, when doing the association study with diplotype (haplotype combination), the effect was shown to be significant, hence only the association analysis between diplotypes and incidence of clinical mastitis was reported in Tables 5 and 6. Moreover, the result indicated that period of calving, and season of calving (S3 Table) were all found to be significant for the incidence of clinical mastitis. The results showed that, the incidence of clinical mastitis was higher for the cows calved during period 5 (2003–2007) and 6 (2008–2012) than for cows calved during period 7 (2013–2020), with odds ratios of 4.41 and 4.61 for period 5 and 6, respectively, compared to period 7 (S3 Table). Similarly, cows that calved between April and June (season 2) had higher incidence of clinical mastitis, with an odds ratio of 8.87 in season 2 (April-June) compared to season 1 (December-March), season 3 (July-September), and season 4 (October-November) (S3 Table).

Logistic regression analysis of haplotype combination in promoter variants of LAP3 gene revealed that, cows carrying haplotype combination H1H1 and H1H4 were more prone to clinical mastitis than other haplotype combinations, with odds ratios of 5.53 (1.16–26.38) and 10.36 (1.79–59.82), respectively (Table 6). Whereas, cows with the haplotype combinations H1H2 and H1H3 were shown to be less susceptible to the incidence of clinical mastitis than other cows, as the odds ratio for the non-incidence of clinical mastitis was found to be lower (Table 6). This was clearly confirmed in the $\chi^2$ values given in Table 5, the frequency of haplotype combination H1H2 (34.78%) and H1H3 (28.00%) in the mastitis affected animal group was low. In this study, cows with haplotype combination H1H3 had higher EBVs for LMY, 305dMY and LL (Table 4) as well as being resistant to clinical mastitis (Tables 5 & 6). As a result, inclusion of these markers in the selection program will speed up the simultaneous improvement of mastitis resistance and milk production traits and thus improving the dairy farm profitability. Looking at the effect of haplotype combinations rather than individual SNPs,

**Table 5. Chi–square values for mastitis affected and not affected animals with respect to genetic variants of LAP3 gene in Sahiwal and Karan Fries cattle.**

| Diplotypes | | Mastitis | | | | Total | $\chi^2$ value |
|---|---|---|---|---|---|---|---|
| | | Non-affected animals | | Affected animals | | | |
| | | No | (%) | No | (%) | | 3.55 |
| H1H1 (CC-TT-AA-CC-GG-CC-TT) | | 36 | 62.07 | 22 | 37.93 | 58 | |
| H1H2 (CC-TT-AG-CC-GG-CT-TT) | | 15 | 65.22 | 8 | 34.78 | 23 | |
| H1H3 (CG-TC-AG-CT-GA-CC-TG) | | 18 | 72.00 | 7 | 28.00 | 25 | |
| H1H4 (CC-TT-AG-CC-GG-CC-TT) | | 12 | 48.00 | 13 | 52.00 | 25 | |
| H3H4 (CG-TC-GG-CT-GA-CC-TG) | | 11 | 68.75 | 5 | 31.25 | 16 | |

No = Number of observations, $\chi^2$ = Chi–square

**Table 6. Effect of diplotypes of LAP3 gene on the incidence of clinical mastitis in Sahiwal and Karan Fries cattle.**

| Effect | Wald chi square | Odds ratio | 95% CI |
|---|---|---|---|
| Diplotype* H1H1 | 5.39 | 5.53 | 1.16–26.38 |
| H1H2 | 1.85 | 1.08 | 0.19–6.18 |
| H1H3 | 2.96 | 0.990 | 0.19–5.22 |
| H1H4 | 9.36 | 10.36 | 1.79–59.82 |
| H3H4 | - | - | - |

* = P < 0.05, CI = Confidence interval, SE = Standard error

it is found that using haplotype combinations within the promoter variants of LAP3 gene gives a better indication of a cow's ability to resist incidence of clinical mastitis and improved lactation performance, and could resolve the prior studies' conflicting association results focusing on a single SNP. The genotype combination effect is a reflection of interactions between many SNPs, hence genotype combination analysis is preferred to single SNP analysis [20]. This agrees with [43], who found that genotype combinations were more effective in terms of inheritance than a single SNP. The area under the curve (AUC), also known as the index of accuracy (A) or concordance (c) statistics corresponding to the ROC curve, is a standard measure of the logistic regression model's predictive accuracy when the outcomes are binary [44]. The higher the area under the curve, the better model's prediction power. In the present study the model's concordance (c) value is 0.79, indicating that the model is more accurate.

## Prediction of TFBS changes caused by the SNPs and investigation of CpG islands in LAP3 promoter region

By performing MatInspector in the genomatix suite, three SNPs in the promoter region of the LAP3 gene, rs720373055: T>C, rs715189731: A>G, and rs720349928: G>A, altered the binding sites of some transcription factors (Fig 2). The substitution of T to C in the rs720373055: T>C and A to G in the rs715189731: A>G resulted in the generation of TFBS for C2H2 zinc finger transcription factor 26 (ZF26). The allele G of rs720349928: G>A created TFBSs for ZTRE (Zinc transcriptional regulatory element), CTCF and BORIS gene family (CTCF+-BORIS), AHRR (AHR-arnt heterodimers and AHR-related factors), ZF15 (C2H2 zinc finger transcription factors 15) and HESF (vertebrate homologues of enhancer of split complex) that were eliminated upon its substitution by the A allele (Fig 2).

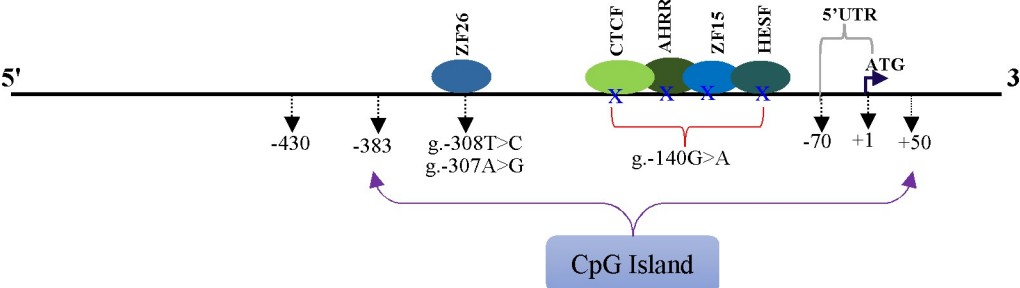

**Fig 2. Bioinformatics analysis and schematic illustration of SNPs that altered transcription factor binding sites in promoter region of the bovine LAP3 gene.** The translation start site (ATG) is marked with +1, the transcription factor binding sites are shown with ellipse, g.– 308T>C (rs720373055: T>C), g.– 307A>G (rs715189731: A>G) and g.–140G>A (rs720349928: G>A). A blue X means transcription factor binding sites eliminated up on its substitution by the mutant allele.

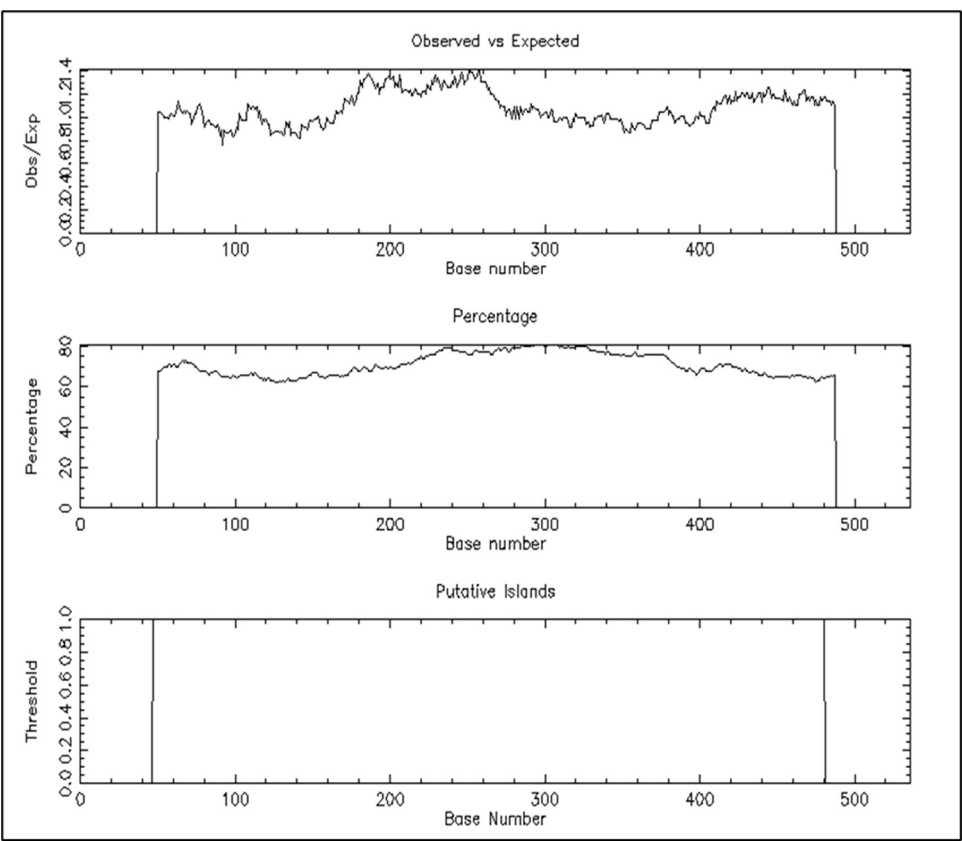

**Fig 3. Promoter CpG island analysis in promoter region of the bovine LAP3 gene by the EMBOSS CpG plot (http://www.ebi.ac.uk/Tools/seqstats/emboss_cpgplot/) with a window size of 100 bp and the following set options: Observed CpG/Expected CpG ratio ≥ 0.65, GC content > 50% and length > 200 bp.** Only one CpG island (433 bp) was detected within the sequence –48 to –481.

To further explore the regulatory elements that are involved in the cattle LAP3 promoter region, CpG islands were investigated using two algorithms. Using the CpG island searcher program (CpGi130) accessible at web link http://dbcat.cgm.ntu.edu.tw/ and the EMBOSS CpG plot (https://www.ebi.ac.uk/Tools/seqstats/emboss_cpgplot/), it was found one (1) CpG island located within the region of -48 to -481 bp in the promoter region (Fig 3). These findings are comparable with those of [23], who found a CpG island in the 5' flanking region (-448 bp to +55 bp) of the bovine PEPS (LAP3) gene and a core promoter area in the -587 bp to -236 bp region, as well as a typical TATA box from the translation start codon. Polymorphisms that influence gene transcription and expression are gaining a lot of interest since they are responsible for a lot of heritable phenotypic differences [45].

## Conclusion

In conclusion, it first explored genetic variants in the promoter and 5'UTR of the LAP3 gene and uncovered associations between the SNPs and milk production traits as well as clinical mastitis in Sahiwal and Karan Fries cattle. The results demonstrated that the SNPs rs720373055: T>C and rs720349928: G>A were significantly associated with EBVs of LMY and 305dMY by possibly influencing TFBs, and that rs722359733: C>T polymorphism was significantly related to LL. In addition, the H1H3 (CTACGCT/ GCGTACG) diplotype had significantly higher estimated breeding values of LMY, 305dMY, and LL than other diplotypes.

Cows with H1H2 and H1H3 haplotype combinations were found to be less susceptible to the incidence of clinical mastitis than other cows. Selection of H1H3 diplotyped animals will therefore accelerate genetic progress in the positive direction for both milk production and mastitis resistance, with an aid to selection for simultaneous improvement of animals. However, further research into the validation of these associations in larger population, as well as functional validation, is required before they can be incorporated into a panel for MAS program. If further research confirms our hypothesis, the H1H3 diplotype could be a convincing molecular marker in breeding dairy cows for high milk production and high mastitis resistance.

## Supporting information

**S1 Table. Haplotypes and haplotype frequencies of seven SNPs in promoter region of the bovine LAP3 gene promoter region.** H = Haplotypes.
(DOCX)

**S2 Table. Allele substitution effect in rs720373055: T>C and rs720349928: G>A polymorphisms in promoter region of the bovine LAP3 gene on lactation milk yield and 305-day milk yield.** [a,b]Means with different superscripts are significantly different ($P < 0.05$); $\alpha$ = Allele substitution effect.
(DOCX)

**S3 Table. Effect of breed, period of calving and season of calving on the incidence of clinical mastitis in Sahiwal and Karan Fries cattle.** ** = $P < 0.01$; * = $P < 0.05$; CI = Confidence interval; P/calving = Period of calving; S/calving = Season of calving.
(DOCX)

## Acknowledgments

Authors are thankful to the Director, ICAR- National Dairy Research Institute, Head, Dairy Cattle Breeding Division for providing necessary facilities in the lab for carrying out the above work.

## Author Contributions

**Conceptualization:** Destaw Worku, Gopal Gowane, Archana Verma.

**Data curation:** Destaw Worku.

**Formal analysis:** Destaw Worku.

**Funding acquisition:** Archana Verma.

**Investigation:** Destaw Worku.

**Methodology:** Destaw Worku.

**Project administration:** Archana Verma.

**Resources:** Gopal Gowane, Archana Verma.

**Software:** Destaw Worku.

**Supervision:** Gopal Gowane, Archana Verma.

**Validation:** Destaw Worku, Gopal Gowane, Archana Verma.

**Visualization:** Destaw Worku, Gopal Gowane, Archana Verma.

**Writing – original draft:** Destaw Worku.

**Writing – review & editing:** Destaw Worku, Gopal Gowane, Archana Verma.

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
