## [Decision Letter · Decision Letter 0]

5 Jul 2022

PONE-D-22-15483Genetic variation in promoter region of the bovine LAP3 gene associated with estimated breeding values of milk production traits and clinical mastitis in dairy cattlePLOS ONE

Dear Dr. Worku,

Thank you for submitting your manuscript to PLOS ONE. After careful consideration, we feel that it has merit but does not fully meet PLOS ONE’s publication criteria as it currently stands. Therefore, we invite you to submit a revised version of the manuscript that addresses the points raised during the review process.

Dear Sir, there are the reviewer's comments. Please make the adjustment (in highlight).

Thanks,

Best regards,

We look forward to receiving your revised manuscript.

Kind regards,

Julio Cesar de Souza, Ph.D.

Academic Editor

PLOS ONE

Journal Requirements:

Additional Editor Comments:

Dear Sir, there are the reviewer's comments. Please make the adjustment (in highlight).

Thanks,

Best regards,

Reviewers' comments:

Reviewer's Responses to Questions

**Comments to the Author**

1. Is the manuscript technically sound, and do the data support the conclusions?

Reviewer #1: Partly

Reviewer #2: Yes

2. Has the statistical analysis been performed appropriately and rigorously? 

Reviewer #1: Yes

Reviewer #2: Yes

3. Have the authors made all data underlying the findings in their manuscript fully available?

Reviewer #1: Yes

Reviewer #2: Yes

4. Is the manuscript presented in an intelligible fashion and written in standard English?

Reviewer #1: No

Reviewer #2: Yes

5. Review Comments to the Author

Reviewer #1: Reviewer comments:

Point1: This paper reported the genetic variation of LAP3 gene and 5'UTR promoter region, and uncovered the association between SNP and milk production characteristics and clinical mastitis of Sahiwal and Karan Fries cattle.

Although the data is novel and the results are interesting. However, this paper is generally good written but it can be improved, and some amendments should be addressed.

Point2: the materials and methods have been well written. Author used Primer3 to design Primer; however, author should include the References (lines 137,151, 154, 162, 165, 185 and 189).

Point3: Results and Discussion. The authors should split the combined Result and Discussion part into two parts (Results and Discussion, respectively). By doing so, readers could appreciate the meaning of your work. In many cases, the results of the paper end up with something like "consistent with previous reports", which could not highlight the importance of your work.

Point4: Line 214 no need this description here without figure, the description should be present on the figure.

Point5: In the FIGUERs. All the figures should be titled and described as well.

Point6: in conclusion. "The conclusions seem to be descriptive". Try to summarize the findings you write in the conclusion by limiting reporting to the most important findings.

Reviewer #2: In this manuscript, the authors did a welcome attempt to describe the genetic variation in the promoter region of the bovine LAP3 gene associated with estimated breeding values of milk production traits and clinical mastitis in dairy cattle. The manuscript is interesting and has valuable results. However, the background needs more information regarding the breeds used in this study (which one is indigenous and which one is crossbred?). The authors repeated the use of subjective terms such as "we believe, our result" which is not acceptable in paper writing.

The numbers below correspond to the line numbers where corrections need to be carried out:

103: rephrase “we believe” as “it is believed”. No subjective terms

116: remove “The current study … Karan Fries” and start with Blood samples …

137: rephrase “PCR reaction … GTCTGC-3’) as “Primers LAP3- F (5’- GCTACGTGCAACCTTTCTCC- 3’) and LAP3- R (5’CTCACCTTCGTCAT 141 GTCTGC-3’) were used to amplify 537 bp of the bovine LAP3 gene (430 bp of 5’ flanking region +107 bp of exon 1)” and remove line 138, 139 and 140.

142- 143: please write the PCR components as concentrations and not as volumes.

144: please specify the annealing temperature used in this study

145-146: rephrase “Product specificity … electrophoresis” as “The PCR products were evaluated by 1.5 % agarose gel electrophoresis by staining with ethidium bromide”

149: please rephrase “sent for sequencing” as “sequenced”

150: remove sequencer

160: remove “for each polymorphism” and replace “analysed” with “calculated”

165: replace “SAS 9.2” with “SAS Version 9.2”

180: replace “LOGISTIC) with “Logistic”

206: rephrase “we came to the conclusion that” as “ it might be concluded that”

208: Among eleven SNPs being detected, seven of them were chosen for association analysis based on their heterozygosities in both breeds. What is the threshold of the heterozygosities?

214: In Fig 1 there are only 9 SNPs instead of 11???

219: rephrase “corresponding χ 2 test to assess whether or not the population was in Hardy-Weinberg equilibrium” as “Hardy–Weinberg equilibrium χ 2 test for the identified SNPs of LAP3 gene promoter variants in Sahiwal and Karan Fries cattle”

234: Table 2 suppose to summarise the genotypic and allelic frequencies for all the eleven SNPs.

261: rephrase “we found” as “it was found”

279: rephrase “we further… cattle” as “the effects of haplotype combinations on EBVs of milk production traits in cattle was analyzed”

287: rephrase “In this study, we show for the first time” as “This study shows for the first time”

290- 299: should be moved to the introduction.

309: rephrase “we believe that” as “ it is believed that”

347: rephrase “our result indicated” as “ the result indicated”

367: rephrase “we found” as “it is found that”

378: in Table 5, Chi-square values are missing.

399: rephrase “we found one” as “it was found one…”

400: rephrase “Our findings” as “these findings”

417: rephrase “In conclusion, we first explored” as “In conclusion, it first explored…”

6. PLOS authors have the option to publish the peer review history of their article (what does this mean?). If published, this will include your full peer review and any attached files.

Reviewer #1: **Yes: **Adam Abied

Reviewer #2: No

---

## [Author Response · Author response to Decision Letter 0]

15 Jul 2022

Academic Editor

PLOS ONE

Dear Julio Cesar de Souza, Ph.D.

We would like to express our great appreciation to academic editor and the reviewers for a thorough reading and constructive criticism of our manuscript entitled “Genetic variation in promoter region of the bovine LAP3 gene associated with estimated breeding values of milk production traits and clinical mastitis in dairy cattle’’ (Manuscript ID: PONE-D-22-15483) and for the opportunity to revise and resubmit. All of these comments were very helpful for revising and improving our paper. We have considered these comments carefully and made corresponding corrections that we hope will meet with your approval. We are pleased to submit the revised research article. The changes in the revised manuscript are highlighted in blue, purple and red. On the following pages, you will find our response to the editor and reviewer comments.

On behalf of my co-authors, I thank you for your consideration of this resubmission. We appreciate your time and look forward to your response. 

Sincerely,

Destaw Worku, PhD (corresponding author)

Animal Genetics and Breeding, 

Salale University, Salale, Ethiopia

Authors response to the comments of the editor and reviewers for the manuscript “Genetic variation in promoter region of the bovine LAP3 gene associated with estimated breeding values of milk production traits and clinical mastitis in dairy cattle”

Journal Requirements:

1. Please ensure that your manuscript meets PLOS ONE's style requirements, including those for file naming. The PLOS ONE style templates can be found at: 

Response: Authors are thankful for providing us this link of PLOS ONE style requirements. According to your suggestion, we have carefully revised our manuscript and addressed PLOS ONE’S style requirements including manuscript body formatting guidelines. We believe that now the manuscript is in a better format and we hope that the revised manuscript fully adheres PLOS ONE's style requirements.

Response: We would like to thank the reviewer for this observation and hence we have included full ethics statement under the materials and methods section of revised manuscript (Line number 123-128).

Reviewer(s) Comments to the Author

Reviewer #1: 

Point1: This paper reported the genetic variation of LAP3 gene and 5’UTR promoter region, and uncovered the association between SNP and milk production characteristics and clinical mastitis of Sahiwal and Karan Fries cattle. Although the data is novel and the results are interesting. However, this paper is generally good written but it can be improved, and some amendments can be addressed 

Response: The Author’s would like to thank the reviewer for the detailed review of the manuscript and thoughtful comments. We have revised our manuscript, and made some amendments. We have addressed each point below and indicated in blue font in the revised manuscript. 

Point2: the materials and methods have been written well. Authors used Primer3 to design Primer; however, author should include the References (lines 137,151, 154, 162, 165, 185 and 189).

Response: References are included as requested under the section heading of references in the revised manuscript (Lines 497, 499, 501, 505, 510 and 515).

Point3: Results and Discussion. The authors should split the combined Result and Discussion part into two parts (Results and Discussion, respectively). By doing so, readers could appreciate the meaning of your work. In many cases, the results of the paper end up with something like "consistent with previous reports", which could not highlight the importance of your work.

Response: We thank the reviewer to clarify this point. However, purposefully, we have combined the results and discussion section in this study, so as to allow more coherence and to discuss the results immediately after presenting them. However, as we could sense from your comments, this section is now written with care to avoid distraction of focus. 

Point4: Line 214 no need this description here without figure, the description should be present on the figure.

Response: This was done as per the PLOS ONE journal’s requirement for manuscript submission. While we understand the reviewer's concern regarding the description without figure, we would like to point out that the PLOS ONE journal instruction for manuscript submission requires that figure captions are inserted immediately after the first paragraph in which the figure is cited. Figure files are uploaded separately. However, tables are inserted immediately after the first paragraph in which they are cited without uploading the tables separately. 

Point5: In the FIGURES. All the figures should be titled and described as well.

Response: All figure titles are described well as requested. 

Point6: in conclusion. "The conclusions seem to be descriptive". Try to summarize the findings you write in the conclusion by limiting reporting to the most important findings.

Response: Authors are thankful for the comment and following the comment, we have taken the opportunity to summarize the findings and further improve the conclusions in the revised draft. Please see the conclusion section of the revised draft. 

Reviewer #2:

In this manuscript, the authors did a welcome attempt to describe the genetic variation in the promoter region of the bovine LAP3 gene associated with estimated breeding values of milk production traits and clinical mastitis in dairy cattle. The manuscript is interesting and has valuable results. However, the background needs more information regarding the breeds used in this study (which one is indigenous and which one is crossbred?). The authors repeated the use of subjective terms such as "we believe, our result" which is not acceptable in paper writing.

Response: Dear reviewer, thank you very much for your detailed review of the manuscript and we sincerely appreciate these well-thought comments. The reviewer’s comments are very helpful for improving our manuscript. We have gone through each and every query and addressed all the comments in depth, which we hope to meet with acceptance requirements. 

It is to note that Sahiwal is one of the most important milch breeds of indigenous cattle famous for higher milk production, remarkable power of endurance for hot climate of subtropics, comparatively resistant to diseases and low maintenance cost. Whereas, Karan Fries is a synthetic breed evolved by crossing Holstein Friesian (75%) and Tharparkar cattle. Therefore, the background information regarding the breeds used in this study have been updated under the section ‘introduction’ (Line number 73-80). Moreover, those subjective terms mentioned in the comments are removed and replaced with appropriate terms. All changes are highlighted in purple font in the revised manuscript.

103: rephrase “we believe” as “it is believed”. No subjective terms

Response: Needful done (Line number 111).

116: remove “The current study … Karan Fries” and start with Blood samples …

Response: “The current study involved 220 lactating dairy cows (110 Sahiwal and 110 Karan Fries)” removed and the paragraph started with blood samples…. (Line number 131).

137: rephrase “PCR reaction GTCTGC- 3’) as “Primers LAP3- F (5’- GCTACGTGCAACCTTTCTCC- 3’ and LAP3- R (5’CTCACCTTCGTCAT 141 GTCTGC- 3’) were used to amplify 537 bp of the bovine LAP3 gene (430 bp of 5’ flanking region +107 bp of exon 1)” and remove line 138, 139 and 140.

Response: This is rephrased in the revised manuscript now (Line number 151-153).

142- 143: please write the PCR components as concentrations and not as volumes.

Response: The PCR components re-written as concentrations (Line number 155).

144: Please specify the annealing temperature used in this study

Response: The annealing temperature used in this study is 59.50C. We have now provided the annealing temperature used in the revised draft. Please see line number 157.

145-146: rephrase “Product specificity … electrophoresis” as “The PCR products were evaluated by 1.5 % agarose gel electrophoresis by staining with ethidium bromide” 

Response: We have modified the sentence as required (Line number 157-159).

149: please rephrase “sent for sequencing” as “sequenced” 

Response: Needful done at line number 162.

150: remove sequencer 

Response: Thank for the comment. We have now removed the word sequencer.

160: remove “for each polymorphism” and replace “analysed” with “calculated” 

Response: The needful changes have been made in the revised draft (Line number 173).

165: replace “SAS 9.2” with “SAS Version 9.2”

Response: We have made the required changes in the mentioned line (Line number 177). 

180: replace “LOGISTIC) with “Logistic”

Response: Thank you so much for catching this error, which we have replaced the word “LOGISTIC” with “Logistic” in the revised draft (Line number 192).

206: rephrase “we came to the conclusion that” as “ it might be concluded that”

Response: We agree with the comment and hence we have modified the text accordingly (Line number 219).

208: Among eleven SNPs being detected, seven of them were chosen for association analysis based on their heterozygosities in both breeds. What is the threshold of the heterozygosities?

Response: We thank for this opportunity to clarify this point. The authors would like to note that the genotypes which are present in both Sahiwal and Karan Fries cattle breed, with frequency more than 5% were included for association analysis. However, the SNP genotypes showing frequencies less than 5% in both the studied population were excluded from association analysis. Please see line number 221.

214: In Fig 1 there are only 9 SNPs instead of 11???

Response: We thank the reviewer for this observation; the remaining 2 SNPs have now been included in Fig 1. of the revised draft and the problem has been fixed. Please see Figure 1 of the revised manuscript.

219: rephrase “corresponding χ 2 test to assess whether or not the population was in Hardy-Weinberg equilibrium” as “Hardy–Weinberg equilibrium χ 2 test for the identified SNPs of LAP3 gene promoter variants in Sahiwal and Karan Fries cattle”

Response: We agree that “corresponding χ 2 test to assess whether or not the population was in Hardy-Weinberg equilibrium” has been re-written to “Hardy-Weinberg equilibrium χ2 test for the identified SNPs of LAP3 gene promoter variants in Sahiwal and Karan Fries cattle” (Line number 232-234).

234: Table 2 suppose to summarise the genotypic and allelic frequencies for all the eleven SNPs. 

Response: We thank the reviewer for this important remark, but for this analysis, we respectfully point out that, among eleven SNPs being detected, seven of them, which are present in both the breeds with frequency more than 5% were chosen for association analysis. In view of this, table 2 summarized the genotypic and allelic frequencies for those seven SNPs chosen for association analysis only.

261: rephrase “we found” as “it was found”

Response: Agreed. “we found” replaced with “it was found” (Line number 274).

279: rephrase “we further… cattle” as “the effects of haplotype combinations on EBVs of milk production traits in cattle was analyzed”

Response: Needful done at line number 292-293.

287: rephrase “In this study, we show for the first time” as “This study shows for the first time”

Response: Needful done at line number 300.

290- 299: should be moved to the introduction.

Response: Authors are thankful for the comment and we have now moved this paragraph to the introduction section of the revised draft (Line number 94-103).

309: rephrase “we believe that” as “ it is believed that” 

Response: This is done at line number 342.

347: rephrase “our result indicated” as “ the result indicated” 

Response: This is done at line number 344.

367: rephrase “we found” as “it is found that”

Response: Done at line number 364.

378: in Table 5, Chi-square values are missing.

Response: We thank the reviewer for indicating this missing Chi-square value. But we are not sure about this comment since Chi-square value i.e 3.55 is already indicated in Table 5. We would appreciate if the reviewer could check this missing value in Table 5 (Line number 376-378).

399: rephrase “we found one” as “it was found one…” 

Response: “we found one” replaced with “it was found one” at line number 396.

400: rephrase “Our findings” as “these findings”

Response: Done. “Our findings” replaced with “these findings” at line number 397.

417: rephrase “In conclusion, we first explored” as “In conclusion, it first explored…”

Response: Needful done at line number 414.

---

## [Decision Letter · Decision Letter 1]

20 Sep 2022

PONE-D-22-15483R1Genetic variation in promoter region of the bovine LAP3 gene associated with estimated breeding values of milk production traits and clinical mastitis in dairy cattlePLOS ONE

Dear Dr. Worku,

Thank you for submitting your manuscript to PLOS ONE. After careful consideration, we feel that it has merit but does not fully meet PLOS ONE’s publication criteria as it currently stands. Therefore, we invite you to submit a revised version of the manuscript that addresses the points raised during the review process.

Dear Sir, It is a pleasure to communicate that your work received the suggestions. We would like you to make the adjustments to complete the job.Best regards,Julio Souza

We look forward to receiving your revised manuscript.

Kind regards,

Julio Cesar de Souza, Ph.D.

Academic Editor

PLOS ONE

Additional Editor Comments (if provided):

Dear Sir,

It is a pleasure to communicate that your work received the suggestions. We would like you to make the adjustments to complete the job.

Yours sincerely,

Júlio Souza

Reviewers' comments:

Reviewer's Responses to Questions

**Comments to the Author**

1. If the authors have adequately addressed your comments raised in a previous round of review and you feel that this manuscript is now acceptable for publication, you may indicate that here to bypass the “Comments to the Author” section, enter your conflict of interest statement in the “Confidential to Editor” section, and submit your "Accept" recommendation.

Reviewer #1: All comments have been addressed

Reviewer #2: All comments have been addressed

2. Is the manuscript technically sound, and do the data support the conclusions?

Reviewer #1: Yes

Reviewer #2: Yes

3. Has the statistical analysis been performed appropriately and rigorously? 

Reviewer #1: Yes

Reviewer #2: Yes

4. Have the authors made all data underlying the findings in their manuscript fully available?

Reviewer #1: Yes

Reviewer #2: Yes

5. Is the manuscript presented in an intelligible fashion and written in standard English?

Reviewer #1: Yes

Reviewer #2: Yes

6. Review Comments to the Author

Reviewer #1: The authors have adequately addressed the raised comments; I am satisfied and recommend the manuscript accepted for publication.

Reviewer #2: All required questions have been answered and the authors have responded satisfactorily to my comments.

7. PLOS authors have the option to publish the peer review history of their article (what does this mean?). If published, this will include your full peer review and any attached files.

Reviewer #1: **Yes: **Adam Abied

Reviewer #2: No

---

## [Author Response · Author response to Decision Letter 1]

26 Sep 2022

Review Comments to the Author

Reviewer #1: The authors have adequately addressed the raised comments; I am satisfied and recommend the manuscript accepted for publication.

Response: Authors are thankful for the comment.

Reviewer #2: All required questions have been answered and the authors have responded satisfactorily to my comments.

Response: Authors are thankful for the comment.

Additional minor corrections to the manuscript:

We have added the approval no. to the ethical statement at the materials and methods section of the revised manuscript (line number 128). Moreover, we have included one additional reference to the reference section of revised manuscript (line number 475).

---

## [Editor Report · Decision Letter 2]

21 Oct 2022

Genetic variation in promoter region of the bovine LAP3 gene associated with estimated breeding values of milk production traits and clinical mastitis in dairy cattle

PONE-D-22-15483R2

Dear Dr. Worku,

We’re pleased to inform you that your manuscript has been judged scientifically suitable for publication and will be formally accepted for publication once it meets all outstanding technical requirements.

Kind regards,

Julio Cesar de Souza, Ph.D.

Academic Editor

PLOS ONE

Additional Editor Comments (optional):

Dear sir, whereas reviewers 1 and 2 accepted the adjustments.

I request that the paper be sent for publication.

Julio Souza
---

## [Editor Report · Acceptance letter]

11 May 2023

PONE-D-22-15483R2 

Genetic variation in promoter region of the bovine LAP3 gene associated with estimated breeding values of milk production traits and clinical mastitis in dairy cattle 

Dear Dr. Worku:

I'm pleased to inform you that your manuscript has been deemed suitable for publication in PLOS ONE. Congratulations! Your manuscript is now with our production department. 

Kind regards, 

on behalf of

Dr. Julio Cesar de Souza 

Academic Editor

PLOS ONE